environmental science/hydrology/ecology

wetlandscape, ecohydrology, stochastic modelling, metapopulation, ecological networks

**Author for correspondence:**
L. E. Bertassello
e-mail: lbertass@purdue.edu

# Dynamic spatio-temporal patterns of metapopulation occupancy in patchy habitats

L. E. Bertassello[1], E. Bertuzzo[3], G. Botter[4], J. W. Jawitz[5], A. F. Aubeneau[1], J. T. Hoverman[2], A. Rinaldo[4,6] and P. S. C. Rao[1]

[1]Lyles School of Civil Engineering, and [2]Forestry and Natural Resources, Purdue University, West Lafayette, IN 47907-2051, USA
[3]Dipartimento di Scienze Ambientali, Informatica e Statistica, Università Ca' Foscari Venezia, 30172 Venezia-Mestre, Italy
[4]Department of Civil, Architectural and Environmental Engineering, University of Padua, I-35100 Padua, Italy
[5]Soil and Water Sciences Department, University of Florida, Gainesville, FL 32611, USA
[6]Laboratory of Ecohydrology, École Polytechnique Fédérale de Lausanne, 1015 Lausanne, Switzerland

 LEB, 0000-0001-5168-2142; EB, 0000-0001-5872-0666;
GB, 0000-0003-0576-8847; JWJ, 0000-0002-6745-0765;
AFA, 0000-0001-8373-9342; JTH, 0000-0002-4002-2728;
AR, 0000-0002-2546-9548; PSCR, 0000-0002-5982-3736

Spatio-temporal dynamics in habitat suitability and connectivity among mosaics of heterogeneous wetlands are critical for biological diversity and species persistence in aquatic patchy landscapes. Despite the recognized importance of stochastic hydroclimatic forcing in driving wetlandscape hydrological dynamics, linking such effects to emergent dynamics of metapopulation poses significant challenges. To fill this gap, we propose here a dynamic stochastic patch occupancy model (SPOM), which links parsimonious hydrological and ecological models to simulate spatio-temporal patterns in species occupancy in wetlandscapes. Our work aims to place ecological studies of patchy habitats into a proper hydrologic and climatic framework to improve the knowledge about metapopulation shifts in response to climate-driven changes in wetlandscapes. We applied the dynamic version of the SPOM (D-SPOM) framework in two wetlandscapes in the US with contrasting landscape and climate properties. Our results illustrate that explicit consideration of the temporal dimension proposed in the D-SPOM is important to interpret local- and landscape-scale patterns of habitat suitability and metapopulation occupancy. Our analyses show that spatio-temporal dynamics of patch suitability and accessibility, driven by the stochasticity in hydroclimatic forcing, influence metapopulation occupancy and the topological metrics of the emergent wetlandscape

dispersal network. D-SPOM simulations also reveal that the extinction risk in dynamic wetlandscapes is exacerbated by extended dry periods when suitable habitat decreases, hence limiting successful patch colonization and exacerbating metapopulation extinction risks. The proposed framework is not restricted only to wetland studies but could also be applied to examine metapopulation dynamics in other types of patchy habitats subjected to stochastic external disturbances.

## 1. Introduction

Species dispersal, which is linked to landscape connectivity and habitat heterogeneity, influences the persistence of populations, communities and ecosystem processes [1]. For decades, metapopulation theory has been a critical tool to advance our basic and applied understanding of the importance of dispersal in diverse fields including epidemiology, community ecology, conservation and evolution [2–4]. Although metapopulation theory has provided major insights, there are limitations to the existing theory. In classical metapopulation theory, patch locations and attributes are assumed to be static. Consequently, metapopulation dynamics (e.g. patch occupancy patterns, metapopulation collapse) are driven only by temporally constant patch suitability, connectivity and species traits (e.g. dispersal ability, extinction and colonization rates) [5,6]. However, spatio-temporal dynamics of patch habitat suitability and connectivity are both important for the persistence of individual populations and metapopulations [7,8].

While much research has focused on the spatial dimensions of patch connectivity [6,9,10], also incorporating its temporal component is essential [11]. Dynamic patchy landscapes are characterized by episodic windows of habitat suitability and accessibility. Such transient habitat suitability and connectivity are important at multiple spatial scales (inter- and intra-patch to landscape) and time scales (daily, season and inter-annual). Earlier modelling and monitoring studies have examined patch occupancy in synoptic snapshots of landscapes during different conditions (e.g. wet and dry) or removing a defined percentage of patches from the original habitat to examine patch loss and network fragmentation [12,13]. Such studies are specific to the scenarios, but do not examine continuous temporal dynamics of habitat conditions, linking habitat suitability and connectivity patterns resulting from external stochastic forcing. Thus, understanding the dynamics of the suitability of dispersed patches hosting a focal metapopulation and how temporal variations in connectivity among patches drive species dispersal are still major challenges [14].

Network theory, an integrative approach for evaluating landscape connectivity, is a key tool that can address this challenge [15,16]. Using this framework, habitat patches are represented as nodes and their potential direct connections as links, which are parametrized based on species dispersal abilities. Most natural systems are best described as 'temporal' networks, with dynamic nodes and intermittent links [17], depending on patch attributes. Thus, topological metrics of the node-network vary in time and space, in turn influencing patch accessibility and species dispersal. Network approaches provide spatially explicit but tractable representations of the spatial complexity and dynamics of a patchy habitat landscape [18,19]. The resilience of metapopulations to habitat (i.e. node) loss and the identification of keystone patches critical to landscape connectivity can be assessed using spatio-temporal variations in network topological metrics [20,21]. Such variations, triggered by stochastic external disturbances, impact metapopulations in patchy habitats. For instance, temporal variability of hydroclimatic conditions (e.g. rainfall, temperature) can influence metapopulations inhabiting fragmented landscapes [22,23] with episodic windows of node suitability and connectivity [8].

Because of the complex challenges in monitoring spatial metapopulation dynamics, models are useful for assessing the effects of habitat fragmentation on population occupancy and persistence. One class of metapopulation models which has been applied to practical problems is the stochastic patch occupancy model (SPOM), found to be extremely valuable for estimating metapopulation asymptotic persistence [9,24]. The SPOM approach is based on a static patchy landscape connectivity matrix and thus does not explicitly consider the temporal dynamics of patchy habitats. Here, we hypothesize that static SPOM approaches, based on time-averaged suitability and connectivity patterns, overestimate the probability of metapopulation persistence and underestimate likely collapse in dynamic habitats during persistent adverse conditions.

We expect spatio-temporal variations in patch suitability and accessibility to lead to temporarily unsuitable conditions for species persistence because of the interactions between the time scales of the transient metapopulation dynamics and habitat suitability fluctuations. When such unsuitable conditions persist, decreased patch connectivity limits dispersal, available suitable habitat patches are decreased, and overall species occupancy declines with an increased likelihood of focal species extinction. We further hypothesize that increased landscape connectivity and larger dispersal ability are

manifested in longer dispersal pathways (ecological corridors) and would increase the choices for suitable patches available for species occupancy. However, species with limited dispersal ability would still have limited habitat choices that increase the likelihood of extinction under extreme conditions. To test these hypotheses, we developed a dynamic version of the SPOM (D-SPOM) for time-varying patchy habitats, where the spatio-temporal dynamics of patch attributes (e.g. area, perimeter, gap distances, etc.) driven by stochastic hydroclimatic forcing [25,26], coupled to fixed species attributes (colonization and extinction rates, dispersal ability), determine patch connectivity and occupancy.

As an illustration of iconic patchy habitats, we focus here on wetlandscapes comprising numerous geographically isolated wetlands (GIWs) recognized as hydrological, biogeochemical and ecological hot spots [27–29]. Because wetlands often occur as discrete, heterogeneous patches distributed within a matrix of upland habitat [30,31], many species of wetland-dependent fauna exist as metapopulations that exchange individuals among patches through dispersal [32]. Local populations of wetland species often are small and isolated [33], and thus vulnerable to natural and anthropogenic temporal fluctuations in patch suitability and/or connectivity.

Long-term persistence of metapopulations of wetland species with patch habitats subject to stochastic hydrologic regimes depends on three key factors: (i) total carrying capacity and heterogeneity of the patch habitat landscape; (ii) dispersal of individuals among suitable and accessible aquatic habitat patches, and (iii) a balance of colonization and extinction rates of the focal species. Spatio-temporal variability in these three variables defines the differences between the static and dynamic habitat paradigms, thus metapopulation dynamics in heterogeneous patchy habitats. Intermittent and ephemeral aquatic habitat patches, such as GIWs, may be temporarily (dis)connected because of hydrological dynamics [34–36]. These changes can significantly influence metapopulation persistence and community dynamics [37,38]. Because of such changes, there is a need to address how spatio-temporal variations in patch habitat suitability and connectivity affect metapopulation persistence in dynamic wetlandscapes compared with a traditional static habitat landscape. Moreover, elucidating the effect of potential cross-correlation (synchronicity) between time series of disturbances, network topological metrics and metapopulation dynamics are crucial elements for developing optimal strategies for conservation and management of patchy habitats.

In the following, we present our analyses of metapopulation occupancy dynamics in patch habitats, simulated using D-SPOM, in two contrasting wetlandscapes, one with dense, heterogeneous pothole prairie wetlands in North Dakota, and the other playa lakes in Texas with sparse, homogeneous wetlands. In §2, we describe linking a parsimonious eco-hydrological model with a metapopulation occupancy model (§2.1), the wetland dispersal networks connecting suitable patch habitats (§2.2), and the two wetlandscapes used as case studies (§2.3). We present D-SPOM simulations for temporal dynamics of patch habitat suitability and occupancy, and we compare the results with the static SPOM (§3.1). We evaluate the sensitivity of SPOM and D-SPOM simulations of patch occupancy to species traits (colonization and extinction) and hydroclimatic forcing parameters (rainfall magnitude and frequency) (§3.2). We further examine the spatio-temporal dynamics of the dispersal network connecting occupied patches (§3.3). Finally, we illustrate the importance of explicitly considering dynamics of occupancy and persistence of a hypothetical neutral metacommunity, comprising three metapopulations each occupying specific intra-patch habitat niches (§3.4). In §4, we close with a discussion of the implications of our key findings to wetland management to maintain key habitat patches and dispersal corridors supporting metapopulation persistence.

# 2. Methods

## 2.1. D-SPOM: coupling spatio-temporal dynamics wetland hydrology and metapopulation patch occupancy

We developed a dynamic version of the stochastic patch occupancy model, SPOM [6,39,40], to simulate the presence/absence of a given species in a time-varying wetlandscape. SPOM computes the distribution of occupied patches, in discrete time steps, by considering focal species traits (extinction, colonization and dispersal distance) and patch spatial organization (patch gap distances). A binary state variable, $p_i(t)$, is set to 1 if site $i$ is occupied at time $t$, and 0 otherwise. From an initial distribution of occupied patches, dynamics are modelled as a discrete time Markov chain. At each time step, the model allows unoccupied patches to be colonized by surrounding occupied patches with probability

$$P_{C,i}(t + \Delta t) = P[\, p_i(t + \Delta t) = 1 | p_i(t) = 0].$$ (2.1)

Similarly, species in occupied patches can go extinct with a probability

$$P_{E,i}(t + \Delta t) = P[\, p_i(t + \Delta t) = 0 | p_i(t) = 1]. \tag{2.2}$$

A discrete time SPOM is a homogeneous first-order Markov chain in which the state of the metapopulation at time $t + 1$ depends only on the state (occupancy pattern) of the metapopulation at time $t$ [39]. For each patch and each time step, the probabilities of colonization and extinction events depend on colonization and extinction rates with exponential survival probability [41]

$$P_{C,i}(t) = 1 - \exp(-C_i(t)\Delta t) \tag{2.3}$$

and

$$P_{E,i}(t) = 1 - \exp(-E_i(t)\Delta t), \tag{2.4}$$

where $\Delta t$ is the simulation time step and $C_i(t)$ and $E_i(t)$ are the colonization and extinction rates for the $i$-patch at time $t$. Note that both $C_i(t)$ and $E_i(t)$ are time dependent based on the current conditions of the landscape.

The key difference between the static and dynamic SPOM is that while SPOM assumes static conditions for landscape patches, D-SPOM captures the time variability in habitat structure, embedded by the amount of suitable habitat, $S_i(t)$, and patch spatial configurations, $d_{ij}(t)$. The colonization rate $C_i(t)$ is specified by the following equation:

$$C_i(t) = c \sum_{i \neq j} \exp(-d_{ij}(t)/D) S_j(t) p_j(t), \tag{2.5}$$

where $d_{ij}(t)$ is the pairwise distance between the patches $i$ and $j$, $D$ is the species dispersal distance, $c$ is the species colonization rate and $S_j(t)$ defines patch suitability. The inter-patch distances $d_{ij}(t)$ were estimated based on perimeter-to-perimeter distances, which vary in time with expansion and contraction of patch areas and perimeters. The local extinction rate on the $i$-patch is inversely proportional to the variable $S_i(t)$, i.e. $E_i(t) = e/S_i(t)$.

In classical metapopulation theory [9,42], the metapopulation capacity, $\lambda_{max}$, is derived for a static landscape (i.e. $S_i(t) = S_i$) and captures the impact of landscape structure—the amount of habitat and its spatial configuration—on metapopulation persistence. Species can persist if the equilibrium state $\mathbf{p}_0$ corresponding to global extinction (i.e. characterized by $p_i = 0$ for any $i$) is unstable. This condition is met when the leading eigenvalue of the Jacobian matrix $\mathbf{J}$ of the system linearized around $\mathbf{p}_0$ is positive. By defining a landscape matrix $\mathbf{M}$ consisting of elements $m_{ij} = \exp(-d_{ij}/D) S_i S_j$ for $i \neq j$ and $m_{ii} = 0$, the Jacobian reads $\mathbf{J} = c\mathbf{M} - e\mathbf{I}$, where $\mathbf{I}$ is the identity matrix. The leading eigenvalue of $\mathbf{J}$ can be expressed as $c\lambda_{max} - e$, where $\lambda_{max}$ is the leading eigenvalue of matrix $\mathbf{M}$ that contains information about the landscape and the quality of the patches. Therefore, the persistence condition reads $\lambda_{max} > e/c$ [6,9].

The theoretical prediction of the classical metapopulation theory [6,9] is that a species can persist whenever $\lambda_{max} > e/c$. However, when simulating the stochastic discrete process using SPOM, a species with $\lambda_{max}$ slightly above the threshold $e/c$ could go extinct due to demographic stochasticity. The novelty we introduce here with the D-SPOM consists in calculating the metapopulation capacity $\lambda_{max}$ at each discrete time step because of the different hydrological habitat conditions, $d_{ij}(t)$, $S_i(t)$ and $S_j(t)$ that are manifested at time $t$. In a dynamic landscape, $\lambda_{max}(t)$ represents the survival threshold for the hydrologic conditions at time $t$. Metapopulation dynamics are estimated in terms of wetlandscape occupancy, $\Omega(t)$, which identifies the temporal dynamics of the fraction of patches that are occupied by a focal species.

The advantage of the proposed D-SPOM approach is to explicitly account for patch dynamics influencing habitat availability and accessibility for a focal species. The model requires two fundamental variables of metapopulation dynamics: habitat suitability (e.g. patch areas) and connectivity (e.g. gap distances). The key novelty introduced here is to consider the two quantities as temporally variable, driven by external stochastic forcing. Following Bertassello *et al.* [26], we estimated temporal fluctuations in attributes (e.g. surface area) of each wetland resulting from the net of precipitation falling over each wetland contributing area, evapotranspiration losses and water exchanged with shallow groundwater (see electronic supplementary material for details). The choice of an appropriate proxy for habitat suitability, $S_j(t)$, depends on species characteristics. For example, amphibians may be associated more with wetland perimeters since most anurans and many caudates lay their eggs in shallow water near shorelines [43], while fish are more linked to the stage inside the water body [44].

In this work, $S_j(t)$ is first set equal to wetland area $A_j(t)$, (see §§3.1, 3.2. and 3.3), and in subsequent analyses we consider only partial wetland area, where each wetland is divided into concentric annular

micro-habitats for species that prefer shallow (less than 30 cm), intermediate (30–80 cm) and deep (greater than 80 cm) water depth (see §3.4). In simulating three metapopulations of a hypothetical metacommunity, our key assumptions are that (i) species have particular preferences for each zone, and (ii) they inhabit these zones independent of each other (i.e. no species interactions). By coupling the temporal variability in wetland habitat suitability, $S_j(t)$, and gap distances between wetland habitat patches, $d_{ij}(t)$, with fixed species traits (extinction rate $e$, colonization $c$ and dispersal distance $D$), we used the static SPOM and D-SPOM to generate time series of metapopulation capacity, $\lambda_{\max}(t)$, and patch occupancy, $\Omega(t)$.

## 2.2. Dynamic dispersal networks

Spatio-temporal variations in wetlandscape attributes and connectivity were investigated in parallel by employing a network-based approach. Each wetland is conceived as a node, and a link between a given pair of nodes is established only if the gap distance is less than or equal to a threshold distance defined by the dispersal ability of a given species [16,45]. For each time step, we built the emergent eco-hydrological network by connecting only those wetlands occupied by species that are located within a given threshold distance, $D$. These networks (or series of fragmented networks) represent the eco-hydrological corridors for species to travel across the wetlandscape.

We characterized the structural and functional topology of the eco-hydrological networks using node degree, $k$, node betweenness, $\beta$, and network length, $L$. The former quantifies the number of links incident to a node. From an ecological perspective, patches with lower $k$ are more prone to extinction compared with patches with higher $k$. Node betweenness quantifies the number of times a node acts as a bridge along the shortest path between two other nodes. Nodes with large $\beta$ serve as stepping-stones for species dispersal across the network. To estimate the spatio-temporal variability of the emergent eco-hydrological network, we also used the concept of network length duration curve (NLDC) inspired by the analogous concept of the stream length duration curve in ephemeral river networks [46]. Network length is calculated as the shortest path between the two most distant nodes in the network (i.e. network diameter). The NLDC represents the inverse of the cumulative exceedance probability of the network length and could provide critical information for watershed management by representing the variation in the availability and reliability of the emergent eco-hydrological corridor for the focal species.

## 2.3. Case studies and hydroclimatic data

We tested our framework in two types of US wetlandscapes ($10 \times 10$ km) with contrasting abundance and heterogeneity of isolated wetlands: prairie potholes (North Dakota) and playa lakes (Texas). These landscapes are characterized by different topography as a result of the generating mechanisms that shaped the land surface (e.g. post-glaciation ice sheets receding in N. Dakota versus wind carving in Texas), and different hydroclimatic forcing. Boundaries of the wetland maximum areas were obtained from the National Wetland Inventory (NWI) database. In this analysis, we filtered out from the NWI database all those wetlands classified as riverine, to account only for GIWs. The total wetland area in N. Dakota is much larger ($A_{T,MAX} = 38$ km$^2$) and wetlands are closer together (mean perimeter-to-perimeter distance when wetlands are at maximum size approximately 80 m), compared with the Texas wetlands ($A_{T,MAX} = 5$ km$^2$, mean distance approximately 450 m). The Texas landscape therefore has low wetland density (approx. 1 wetland km$^{-2}$), compared with N. Dakota (approx. 30 wetlands km$^{-2}$).

The dynamic wetlandscape model requires two hydroclimatic inputs: daily rainfall and evapotranspiration. Historical records of daily rainfall data (2000 to 2007) were obtained from the National Oceanic and Atmospheric Administration (NOAA) stations closest (less than 20 km) to the $10 \times 10$ km study areas. Potential evapotranspiration (PET) data were estimated by the Thornthwaite method using temperature data (NOAA stations) at monthly resolution. The full list of other parameters needed to implement the hydrological model can be found in Bertassello *et al.* [26].

# 3. Results

## 3.1. D-SPOM: patch habitat and occupancy dynamics

Simulated time series of total wetland normalized area, $A^*(t)$, defined as the ratio between the total area of all wetlands in the landscape at time $t$ divided by $A_{T,MAX}$, show high variability between annual wet

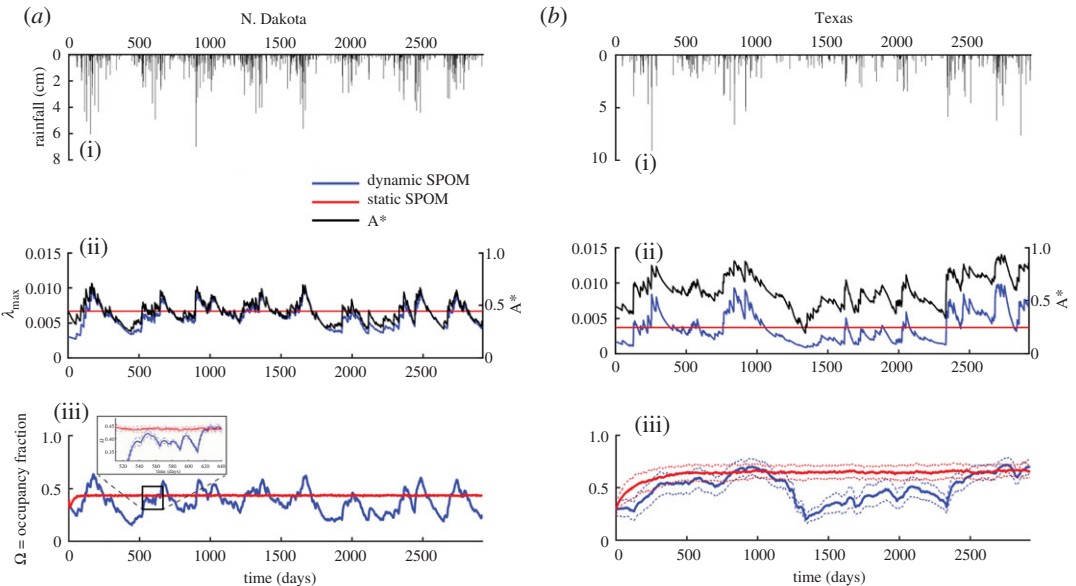

**Figure 1.** Influence of rainfall patterns on the time series of wetlandscape normalized area, metapopulation capacity, $\lambda_{max}(t)$ and the simulated average occupancy, $\Omega(t)$, in two case study wetlandscapes. Simulations with static SPOM are shown for comparison (red lines). Hyetographs (grey histograms) are based on rainfall records. Species attributes used in these simulations are the same in both wetlandscapes: $e = 0.0002$, $c = 1.5$, $D/NND = 1.5$. To account for the inherent stochasticity associated with the static (SPOM) and its dynamic versions (D-SPOM), we ran both the models 100 times. The results are presented in terms of the average occupancy over each single daily realization (solid lines) plus and minus one standard deviation (dashed lines). Note that $\lambda_{max}$ is not affected by this stochasticity since it is only a function of the current hydrological conditions.

**Table 1.** Summary of temporal statistics for wetlandscape normalized area, $A^*(t)$, and mean nearest neighbour separation distances, $NND(t)$, and correlations with metapopulation. Correlation coefficients of the aforementioned variable with metapopulation capacity, $\lambda_{max}(t)$ and average occupancy, $\Omega(t)$.

|  | $\mu \pm \sigma(A^*)$ | $CV(A^*)$ | $\rho_{A^*,\lambda_{max}}$ | $\rho_{A^*,\Omega}$ | $\mu \pm \sigma(NND)$ | $CV(NND)$ | $\rho_{NND,\lambda_{max}}$ | $\rho_{NND,\Omega}$ |
|---|---|---|---|---|---|---|---|---|
| N. Dakota | $0.44 \pm 0.09$ | 0.21 | 0.96 | 0.92 | $85 \pm 10$ m | 0.11 | $-0.82$ | $-0.83$ |
| Texas | $0.56 \pm 0.12$ | 0.22 | 0.87 | 0.84 | $450 \pm 24$ m | 0.05 | $-0.83$ | $-0.82$ |

and dry seasons (figure 1). The mean wetlandscape normalized area, $\mu(A^*)$, is larger in Texas than in N. Dakota, but temporal variability expressed by the coefficient of variation, CV, of $A^*(t)$ is similar in the two wetlandscapes (table 1). The mean nearest neighbour gap distance, NND, in the sparse Texas playa lakes region is about five times larger compared with the dense N. Dakota prairie wetlands. The CV of mean NND, driven by wetland area dynamics, is low in both wetlandscapes, indicating that the changes in wetland diameters are small compared with the wetland separation distances, especially in Texas. The heterogeneity of available habitat also plays a fundamental role in defining the landscape suitability. To single out this contribution, we computed the spatial statistics of wetland area and gap distance at each time step and we compared the ratio, $r_{CV}$, of their CVs. For both wetlandscapes $r_{CV} > 1$, meaning that the contribution of area variability is larger than the gap variability in shaping landscape suitability. In addition, this contribution is much higher in N. Dakota ($\langle r_{CV} \rangle = 4.5$) compared with Texas ($\langle r_{CV} \rangle = 1.3$).

Given the temporal hydrological variability of wetland attributes (e.g. $A^*(t)$), we simulated metapopulation dynamics using the D-SPOM. For every stochastic jump induced in $A^*(t)$ by rainfall events, $\lambda_{max}(t)$ and $\Omega(t)$ increase as well, followed by the recession (figure 1), both due to (i) gradual changes in habitat suitability (decrease in wetland area from evaporation and leakage to upland), and (ii) increases in gap distances, $d_{ij}(t)$, with decreasing area and perimeter. As expected from the $r_{CV}$, the strong correlation of $\lambda_{max}(t)$ with $A^*(t)$ indicates that climate-induced fluctuations in wetlandscape area dominate over the corresponding changes in separation distance defined by wetlandscape spatial configuration. These effects are exacerbated in N. Dakota, where the relative importance of area variability, $r_{CV}$, is large. A similar degree of correlation (table 1) is also observed for the average

occupancy, $\Omega(t)$, suggesting that its pattern is primarily driven by stochastic hydroclimatic forcing (e.g. rainfall) reflected in wetland attributes fluctuations. The correlation between $A^*(t)$ and $\Omega(t)$ is weaker than the correlation between $A^*(t)$ and $\lambda_{\max}(t)$ because $\Omega(t)$ is also affected by the stochastic processes of colonization and extinction, while $\lambda_{\max}(t)$ is mainly a representation of the amount of habitat and its spatial configuration.

The stochasticity of the D-SPOM has different effects on the average occupancy observed in the two wetlandscapes. To analyse this effect, we computed a daily-based standard deviation among all the single realizations of $\Omega(t)$ and we reported its temporal trend as dashed lines in figure 1. While in Texas the standard deviation band is $\sigma_\Omega = \pm 0.1$, in N. Dakota it is close to zero ($\sigma_\Omega < 0.05$). The stochastic fluctuations increase with decreasing number of habitat patches, thus extinction-colonization stochasticity could increase the extinction risk, especially in a less abundant wetlandscape (Texas).

We also compared D-SPOM simulations with static SPOM by fixing wetland areas constant and equal to their mean values. Both the static and dynamic models produce the same mean occupancy, but D-SPOM reveals the temporal dynamics of occupancy, which are critical for understanding metapopulation dynamics and the persistence of several species. This is especially clear in N. Dakota, where the static SPOM severely underestimates the temporal fluctuations in $\Omega(t)$. In Texas, the static SPOM results are similar to D-SPOM during the first three years, but during an extended dry period, a clear discrepancy between the static and dynamic SPOM is evident. While in the static SPOM the average occupancy, $\Omega(t)$, is almost constant, in D-SPOM occupancy decreases from approximately 0.65 to approximately 0.25, because of contraction of wetland areas driven by persistent arid conditions. At the end of this dry period, $\Omega(t)$ gradually recovers over a 3-year period to the original value; thus, in this scenario, the metapopulation does not collapse because of a combination of habitat complexity and species traits ($e$, $c$, $D$). For example, if the focal species was characterized by larger extinction rate, $e$, or smaller $D$, the chances of extinction during the prolonged dry period would have been larger.

## 3.2. D-SPOM sensitivity to species traits and hydroclimatic forcing

We examined the sensitivity of the fraction of surviving metapopulations, $f_S$, first to the variation in species traits ($e$, $c$) and then to the hydroclimatic parameters (mean rainfall depth, $\alpha$, and mean rainfall frequency, $\lambda$) in both the SPOM and D-SPOM (figure 2). For each combination of parameters, we simulated 100 realizations of both the SPOM and D-SPOM and evaluated $f_S$ to assess persistence. The sensitivity of $f_S$ showed similar patterns in the two wetlandscapes, and here we present only the results obtained in N. Dakota.

The ratio of extinction and colonization rates is the derivative of data in figure 2, or the slope of linear relationships. We find that for the static SPOM, the median value of $f_S$ that defines the threshold between extinction and persistence has slope $e/c \sim 10^{-3}$. The SPOM and D-SPOM coincide for small values of extinction, $e$, and colonization, $c$. In these cases, the hydroclimatic variability does not influence the temporal patterns of the metapopulations, which is only affected by the initial species distribution. Indeed, when $e$ and $c$ tend to zero, metapopulation occupancy tends to remain constant because the focal species cannot go extinct and neither colonize new patches. As $e$ and $c$ values increase, differences between SPOM and D-SPOM are evident in the decrease of the slope of the contour lines ($e/c \sim 0.6 \times 10^{-3}$). Thus, for fixed values of $e$ and $c$, the fraction of surviving metapopulations under dynamic conditions is more sensitive for collapse than under steady conditions.

The hydroclimatic parameters ($\alpha$, $\lambda$) also have considerable effects on metapopulation persistence (figure 2). Simulations with SPOM and D-SPOM converge for arid conditions (small $\alpha$ and $\lambda$ values), precluding metapopulation survival. The static and dynamic approaches also converge for wet conditions (large $\alpha$ and $\lambda$ values) in which wetlands are full and large habitat capacity is maintained. Differences between SPOM and D-SPOM are more distinct for intermediate conditions between these end members, where the contour band shifts from the lower left to the upper right corner moving from static to dynamic SPOM. This effect is strong in N. Dakota, with increased sensitivity to variations in $\lambda$ when $\alpha < 0.80$ cm. This trend suggests that as rainfall inter-arrival time increases, species dispersal and survival is compromised during prolonged dry periods.

## 3.3. Spatio-temporal dynamics of dispersal networks

In both SPOM and D-SPOM, patch occupancy dynamics are determined by patch attributes (suitability; gap distances) and species attributes (dispersal ability, colonization and extinction rates).

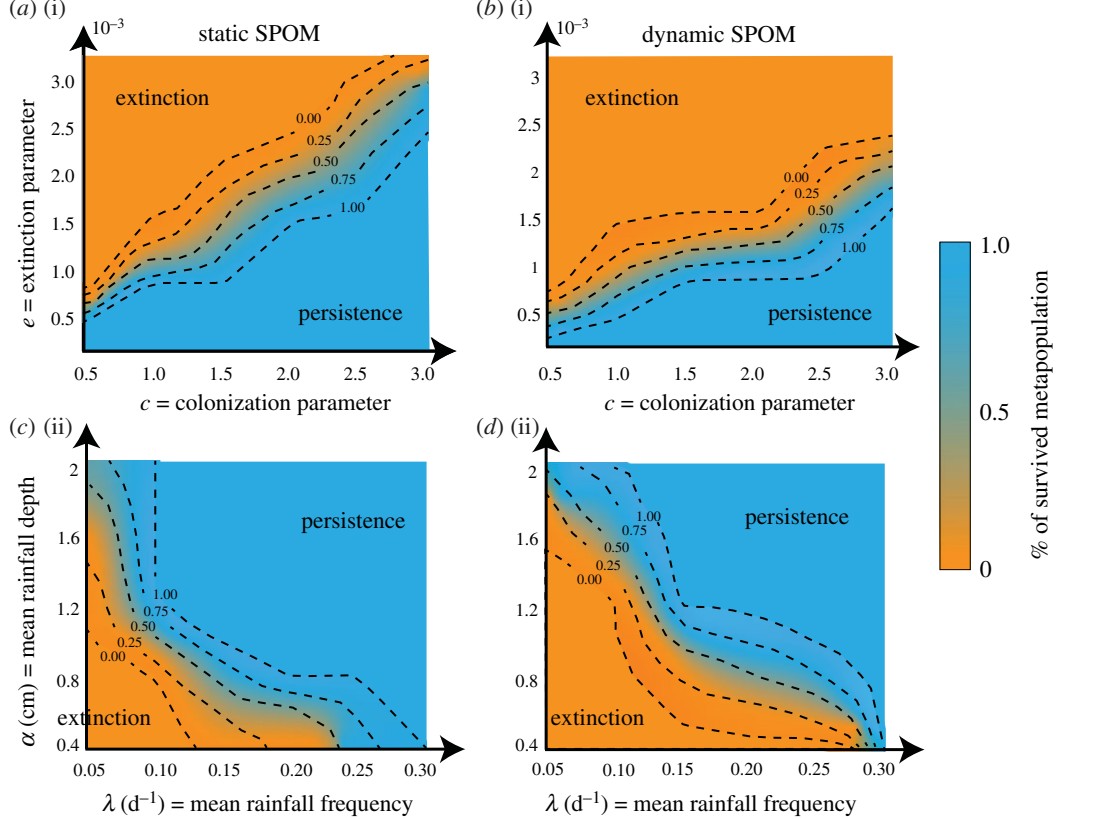

**Figure 2.** Sensitivity of static and dynamic SPOM simulations to changes in species traits ($e$, $c$) and hydrological variability ($\alpha$, $\lambda$) on metapopulation persistence in N. Dakota (see electronic supplementary material, Information for the sensitivity analysis results obtained in Texas). (*a*) Results are shown for sensitivity to species traits, $e$ [0.0001;0.0032] and $c$ [0.5;3.0], with fixed ratio $D/NND = 1.5$, and hydroclimatic parameters ($\alpha = 0.90$ cm, $\lambda = 0.20$ d$^{-1}$ and $ET = 0.50$ cm/d). (*b*) Sensitivity of static and dynamic SPOM to hydroclimatic parameters $\alpha$ [0.40; 2.00] and $\lambda$ [0.05;0.30], by fixing $ET = 0.50$ cm/d, $e = 0.0008$, $c = 1.5$ and $D/NND = 1.5$. For each set of parameter combinations, 100 Monte Carlo simulations were run.

The spatio-temporal dynamics of the dispersal network topology provide additional useful information about the variability in connectivity between components of a metapopulation.

Figure 3 highlights the results in terms of wetlandscape network dynamics for the N. Dakota prairie potholes region. The temporal trends of node degree, $k(t)$, and node betweenness, $\beta(t)$, are strongly affected by rainfall seasonality. Node degrees, $k(t)$, show the largest correlation coefficient (0.95) with the temporal pattern of wetlandscape normalized area, $A^*(t)$, while the correlation between the latter and node betweenness, $\beta(t)$, is weaker (0.71). Indeed, node betweenness of a single wetland is not only affected by its current hydrological status, but also by the current status of the neighbouring wetlands. Therefore, variability in the hydrological response of the neighbouring wetlands could explain the decrease in the correlation between wetlandscape normalized area and node betweenness. Nevertheless, during wet conditions, both $k(t)$ and $\beta(t)$ increase (figure 3*a*) because $d_{ij}(t)$ decreases as more wetlands fill. Thus, more habitat patches become suitable for species colonization, and the emergent eco-hydrological network of occupied wetlands is more clustered, with high $k(t)$ (figure 3*c,d*). A clear eco-hydrological corridor emerges, with high-$\beta(t)$ nodes acting as 'stepping-stones' for sustaining wetlandscape connectivity. For conditions of intermediate wetness (figure 3*c,d*), $k(t)$ and $\beta(t)$ values decrease, with less clustering of wetlands, and the network fragments into two subgraphs. During dry conditions, the network is fragmented into several subnets, characterized by small $k(t)$ and $\beta(t)$ (figure 3*b–d*).

We also quantified the spatio-temporal variability of the emergent eco-hydrological network by using the network length duration curve, NLDC. First, we compared the variability in the networks obtained using the static and dynamic SPOMs. The NLDC estimated from the static (SPOM) approach shows less variability (CV = 0.24), compared with the dynamic approach (D-SPOM) (CV = 0.53). Indeed, while the mean network length, $L(t)$, is similar (approx. 9 km), the standard deviation is almost double in the dynamic approach (4.2 versus 2.6 km). This tendency is well represented in figure 3*b*, where the variability of network length is quantified by the slope of the NLDC, which is proportional to $L(t)$

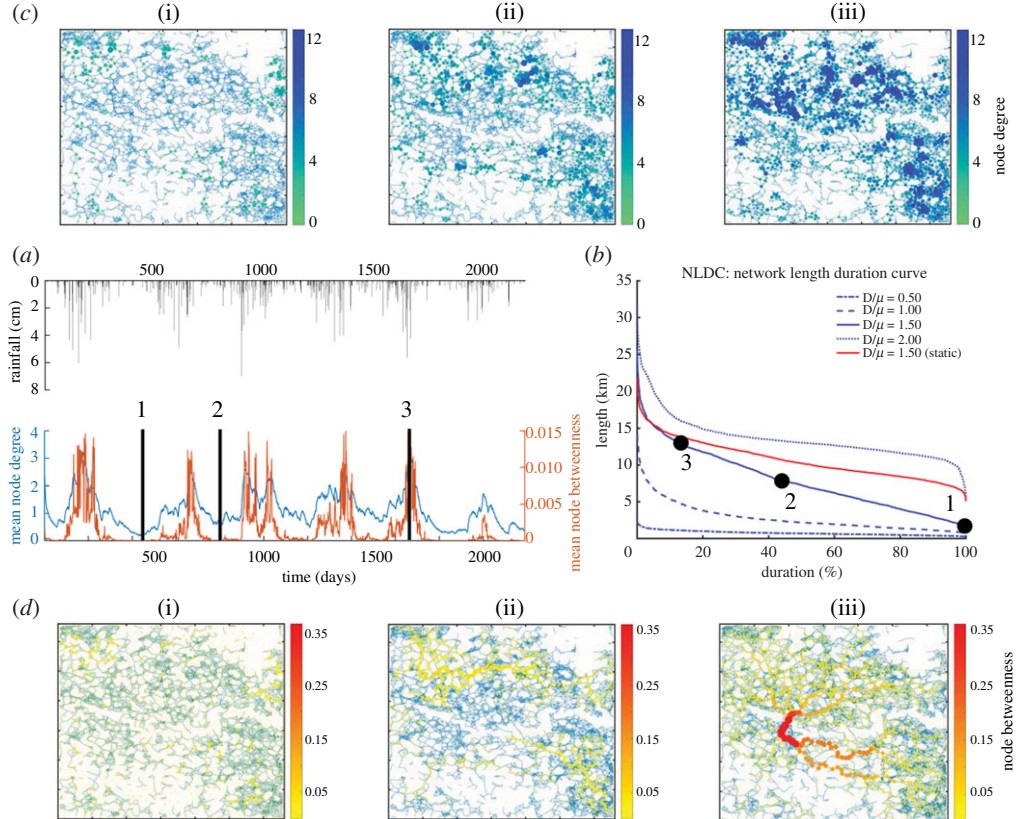

**Figure 3.** (*a*) Spatio-temporal pattern of mean node degree and betweenness in the N. Dakota wetlandscape network. (*b*) Network length duration curves, NLDCs, for different dispersal distances. We also show the map of node degree (*c*) and betweenness (*d*) for three snapshots representing dry (i), intermediate (ii) and wet (iii) conditions.

variance. The difference in the slope of the two NLDCs confirms that the static SPOM cannot adequately reproduce the variability in network length, and in particular, it tends to overestimate the persistence of $L(t)$ when compared with the D-SPOM.

Figure 3*b* also shows the NLDC for three other dispersal distances. The mean network length increases for large dispersal distance because more wetlands can be colonized and thus be connected. However, the variability of $L(t)$ is smaller in all these three new cases compared with $L(t)$ obtained for the base case of $D/NND = 1.5$. This tendency suggests that species with large dispersal distances (e.g. $D/NND > 1.5$) could overcome the effect of the hydrological variability, because even if the closer wetlands are not available, species could access a larger spatial domain, eventually finding suitable habitat. On the other hand, species with smaller dispersal distances (e.g. $D/NND < 1.5$) are limited to small clusters of nearby wetlands. The limited size of these wetland clusters causes on average a shorter network length, $L(t)$ (e.g. $\mu(L) = 2$ km). However, due to the large abundance of these clusters, the temporal variability in $L(t)$ is narrow, resulting in shallow slopes of the respective NLDCs.

## 3.4. Intra-patch heterogeneity and neutral metacommunity dynamics

Heterogeneity of habitat niches within patches (here, wetlands) is an important feature determining species diversity at local (patch) scales and across wetlandscapes. Here, we present an example habitat niche zonation, where each wetland was divided into three habitat niches, whose areas vary in time with fluctuations in rainfall patterns. Each wetland zone is populated by a different species A, B and C independent of each other. Species A is more associated with shallow water near wetland edges, species B to intermediate water and species C to deep water in the wetland centres. In N. Dakota wetlands, the metapopulation capacity $\lambda_{\max}(t)$ is the largest for species C (figure 4) because the portion of habitat provided by the deep water (greater than 80 cm) is much larger compared with the other two zones. Indeed, since the contribution of the exponential term to $\lambda_{\max}(t)$ is the same, the differences between the temporal trends of $\lambda_{\max}(t)$ are entirely related to the sizes of the three zones. However, the mean occupancy, $\langle \Omega \rangle$, for species C is the smallest among the three

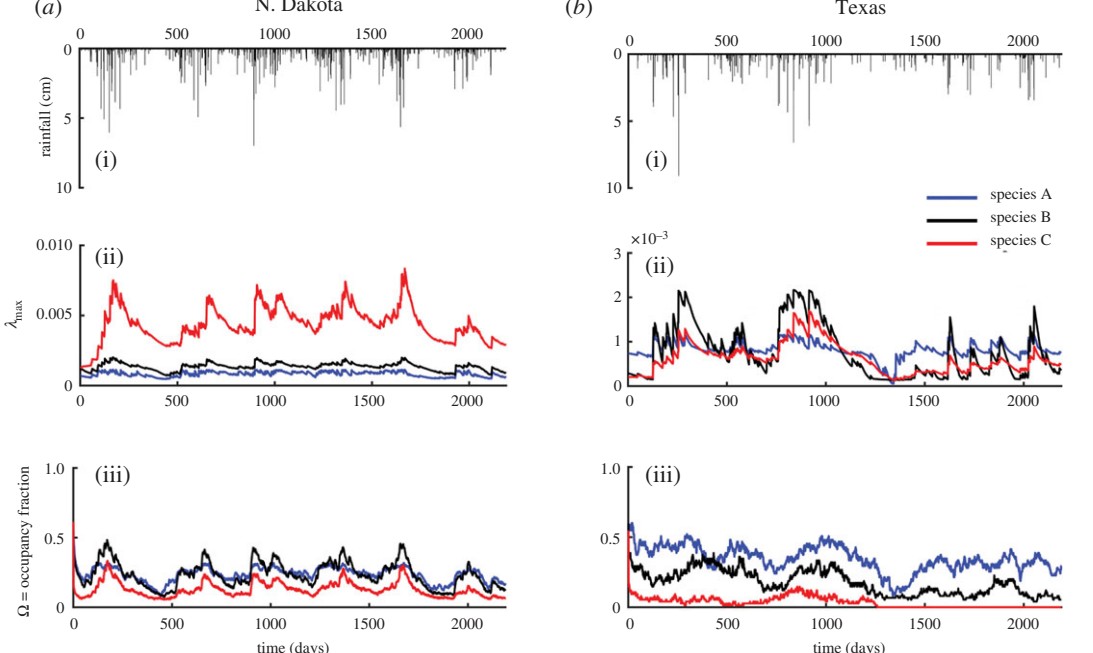

**Figure 4.** Comparison among the time series of metapopulation capacity, $\lambda_{max}(t)$, and occupancy fraction, $\Omega(t)$, for three independent species populating the same wetlandscapes. The three time series for the occupancy fraction represent the mean value of 100 stochastic Monte Carlo simulations. We independently ran the DSPOM model for these three different species in both N. Dakota and Texas and evaluated the metapopulation capacity and the average occupancy for each species.

habitat zones ($\langle\Omega_A\rangle = 0.24$, $\langle\Omega_B\rangle = 0.22$, $\langle\Omega_C\rangle = 0.13$) because its hydroperiod is the shortest, thus reducing species occupancy.

Low variability in $\lambda_{max}(t)$ for Texas playa lakes suggests that the variability in the size of the three micro-habitats is limited. Indeed, while the N. Dakota wetlands are characterized by large bathymetry differences induced by glacial retreat scouring, in the Texas playa lakes the differences in wetland bathymetry and shape are reduced. However, the temporal dynamics of $\Omega(t)$ for the three species shows an important difference. While species A and B survive throughout the entire simulation period, species C goes to extinction after four years of simulation ($t \sim 1200$ days). Indeed, at that time, we observe the longest dry period, which severely impacts water storage dynamics causing a decrease in wetland stage. The deep-water habitat is effectively eliminated during this prolonged dry period, driving species C to extinction.

# 4. Discussion

## 4.1. Key findings

Incorporation of the spatio-temporal variations in patchy habitats is important for modelling the persistence of metapopulations in fragmented and dynamic patchy landscapes. Here, we linked a parsimonious hydrologic model with a well-known patch occupancy model. Spatio-temporal dynamics of patch suitability and connectivity, and thus occupancy, are driven by external stochastic forcing. We compared the outputs of static and dynamic stochastic patch occupancy models (SPOM and D-SPOM) in two contrasting US wetlandscapes. This comparison showed that the extinction risk in dynamic landscapes is exacerbated by extended drought periods when suitable patches are not accessible for successful colonization resulting from dispersal network fragmentation. Examples presented here illustrate how explicit consideration of the temporal dimension in metapopulation approaches is important to interpret local and wetlandscape-scale occupancy patterns in patchy habitats.

Metapopulation dynamics are defined by temporal dynamics of patchy habitat suitability, metapopulation capacity and occupancy. Time-averaging the landscape matrix sequence of patchy habitats, as in the static SPOM, cannot adequately estimate metapopulation temporal dynamics. Interpreting whether metapopulations can cope with chronic small perturbations and recover from infrequent large shocks [11], it requires an understanding of transient habitat conditions and dynamic suitability and connectivity in patchy habitats. Understanding the sensitivity of metapopulation

survival to these perturbations induced by variability in hydroclimatic parameters (figure 2) is also useful to predict their trajectories under climate change scenarios of increased droughts [47]. Temporal variations in intra-patch heterogeneity (i.e. habitat niches for different species), induced by hydroclimatic forcing, also drive metacommunity dynamics, with a decrease in habit niches driving some species to extinction, while other species persist.

The interplay between the species traits ($e$, $c$, $D$), with the spatio-temporal heterogeneity in patchy habitat attributes [$A_i(t)$, $d_{ij}(t)$] and temporal dynamics driven by stochastic hydroclimatic forcing ($\alpha$, $\lambda$, ET), determines the persistence of metapopulations in wetlandscapes. Temporal patterns of recurring conditions for unfavourable patch attributes impact metapopulation persistence, especially in sparse, homogeneous wetlands in arid settings (e.g. Texas) compared with dense and diverse wetlands in more humid conditions (e.g. North Dakota). Large decreases in the overall species occupancy and extinction are more likely when unsuitable conditions and large gap distances persist (e.g. during long-term droughts lasting years or decades). Indeed, a species with dispersal ability that is much larger than the average distance among patches ($D/NND \gg 1$) could access the entire patchy habitat, providing important refugia during times of harsh conditions induced by stochastic hydroclimatic forcing. However, species with smaller $D/NND$ ratios are more susceptible to wetlandscape hydrologic dynamics, and the effects of extinction and colonization are more important.

The network length duration curve, NLDC, represents a flexible and simple concept that can be used to assess the spatio-temporal dynamics of the active fraction of the wetlandscape network, providing a benchmark for characterizing the influence of hydroclimatic variability on a variety of ecological processes (e.g. dispersal, competition and biodiversity). Our results show that the same wetlandscape can provide a multitude of different networks for a given set of species. The variability, and thus persistence, of the emergent eco-hydrological network is characterized by the slope of the NLDC, whose value is impacted by both the wetlandscape spatial configuration and species dispersal abilities. Steeper NLDCs indicate more erratic behaviour for the wetlandscape network, while flatter NLDCs show larger persistence of the emergent ecological corridor. These insights have important implications from a conservation perspective. Indeed, landscape management strategies that aim to increase the persistence of endangered species can identify the preferred locations for new or restored wetlands that allow flattening the slope of the NLDC.

## 4.2. Ecological implications

Dispersal pathways (corridors) used by a focal species to persist in dynamic patch habitat landscapes are identified by D-SPOM, based on dynamic topological metrics of dynamic dispersal networks connecting occupied patches (figure 3). During dry conditions, few wetlands are available for colonization and the network is severely fragmented, impeding species dispersal and occupancy. For each precipitation event, wet conditions are re-established and species can find many suitable patches to colonize. Thus, stochasticity and seasonality of rainfall patterns drive the topology of the dynamic dispersal network. These spatio-temporal patterns are well described by the node degree maps, which show how several clusters emerge with increasing wetness of the landscape. These wetland clusters indicate locations where a group of species with similar life-history traits and dispersal ability congregate [48]. In addition, these clustered wetlands are characterized, on average, by a high local resilience based on supporting a triangular linkage structure [49,50]. On the other hand, node betweenness maps identify wetlands acting as 'stepping-stones' for species dispersal across the domain, thus supporting the identification of ecological corridors used by species to maintain viable metapopulations. Loss or disruption of these wetlands, either from climate change or land-use alteration, decreases available habitats (i.e. metapopulation capacity) and their connectivity (i.e. network fragmentation) and decreases species fitness, thus increasing the risk of metapopulation extinction [32,33]. From a landscape management perspective, it is important to decide the priority of interventions because 'hubs' or 'stepping-stones' wetland do not always coincide (see electronic supplementary material).

Many species use multiple patch habitats, and the spatial complexity of these habitats has strong impacts on population dynamics and long-term viability. Indeed, intra-patch dynamics play an important role in the persistence of species with different preferences for habitat niches. For example, in Texas playa lakes, species that require deeper inundation (e.g. species C: wetland stage greater than 80 cm) cannot find suitable conditions and are driven to extinction (figure 4). In ephemeral wetlands, periodic drying imposes severe constraints on species behaviour, development and life history, such that only species adapted to deal with drying are successful in these habitats [51]. Thus, physical factors constrain habitat availability for species and contribute to their diversity. Determining how spatio-

temporal variability of patchy habitats influences the dispersal of multiple species across the landscape will be critical for investigating metacommunity dynamics and regional biodiversity [52,53].

## 4.3. Advantages and limitations

Our study suggests that predicting species presence and persistence in patchy habitats requires understanding the consequences of stochastic spatio-temporal dynamics of patch suitability and accessibility. Parsimonious hydrologic models were used to generate time series of wetland hydrologic conditions, which were then used to establish temporal dynamics of patch connectivity (network topology). Linking parsimonious hydrological and ecological models in D-SPOM enables prediction of occupancy dynamics in patchy habitats. Here, the linkage is expressed first in terms of wetland area (§3.1) and then by considering a single wetland as divided into three micro-habitats (§3.4). However, depending on the type of focal species, other patch attributes (e.g. wetland perimeter) could be used to define wetland suitability. D-SPOM is a general framework that is not restricted only to wetland studies but could also be applied to other types of patchy habitats whose temporal dynamics are induced by stochastic forcing. For example, hydroclimatic forcing also includes physiologic stress on vegetation resulting from fluctuations in root-zone soil-water storage and the loss of vegetation during persistent droughts and flooded periods. Thus, for insect species dependent on patchy habitats in grasslands, and soil microbial communities critical for nutrient cycling, such stochastic fluctuations drive spatio-temporal dynamics in habitat suitability.

Our analysis does not consider upland matrix effects, where heterogeneity of land cover can either impede or facilitate species dispersal in natural or human-impacted landscapes [54]. Anisotropy in dispersal directions may determine the choice of patch habitat by individuals. Matrix and anisotropy effects can be accommodated by assigning weights to the network links. Non-stationarity in species traits ($c$, $e$, $D$) is also not considered in our analysis, instead represented by fixed average values. This assumption is likely to be violated when evolution modifies species response to environmental variables. Under these circumstances, natural selection may change dispersal rates across an invasive species expanding range [55,56]. Here, we assumed that the metapopulations are adapted to the mean conditions of their respective landscapes. However, when extreme conditions persist, occupancy levels drop significantly, but might slowly recover to 'average' conditions, as evident in the Texas case study simulations.

Finally, difficulties in collecting long-term and high-resolution (e.g. daily-based) monitoring data for individual populations at landscape scale in patchy habitat present challenges for validation of D-SPOM with empirical occupancy data. However, D-SPOM simulations can help identify key patches and critical time periods for hydrologic and species monitoring. For instance, high-degree patch nodes during the wet and dry seasons would be the ideal hotspots for assessing the distribution of focal species in the landscape. Models can also guide habitat management for conservation by identifying those stepping-stones of fundamental importance for enhancing habitat capacity and accessibility, or by using the notion of NLDC to suggest the optimal location for the creation of new habitats.

Data accessibility. The data that support the findings of this study are publicly available. Data for wetland dimension and location are available at the National Wetland Inventory website (https://www.fws.gov/wetlands), while the rainfall data can be found at the National Oceanic and Atmospheric Administration website (https://www.noaa.gov).
Authors' contributions. All authors contributed to the conceptualization and implementation of the model. L.E.B. and P.S.C.R. led the manuscript writing process. All co-authors contributed to expanding the scope of investigations, the revision of manuscript drafts and the final draft submitted for publication.
Competing interests. We declare we have no competing interests.
Funding. P.S.C.R. acknowledges support from two NSF grants: NSF Collaboration Research, RIPS Type 2: Resilience simulation for water, power and road networks, Award no. 1441188; NSF Award no. 1354900, 'Plant adaptation in variable environments'.
Acknowledgements. Lee A. Rieth Endowment in the Lyles School of Civil Engineering at Purdue University provided partial financial support for P.S.C.R. and L.E.B.

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
