## [Reviewer comments · Royal Society Open Science]

Review History

RSOS-201309.R0 (Original submission)

Review form: Reviewer 1

Is the manuscript scientifically sound in its present form?

Yes

Are the interpretations and conclusions justified by the results?

Yes

Is the language acceptable?

Yes

Do you have any ethical concerns with this paper?

No

Have you any concerns about statistical analyses in this paper?

No

Recommendation?

Accept with minor revision (please list in comments)

Comments to the Author(s)

The authors succeeded in doing a great job. I particularly like the description of patch networks by using parameters developed in the framework of network theory. I like very much how the hydrodynamic model is able to link rainfall to network structure in a dynamic way, which ends defining a constantly changing patch network. In sum, I love the combination of metapopulation ecology, network theory and hydrology.

My comment will be minor as I think the paper could be published almost as it is.

When you present the calculation the metapopulation capacity, I am a little bit worried about the interpretation of this parameter in the context of a rainfall-driven patch network. This parameter arises when trying to find a feasible non-trivial solution for the probability vector $p=(p_1, \dots, p_N)$, where p_i is the occupancy probability of the species on patch i . However, you say it is the leading Eigenvalue of the Jacobian of the system, but you don't present clearly enough the equations describing your dynamics. Could you specify the system of equations this Jacobian is representing and at which point is the Jacobian being evaluated? My worry is that if you defined the metapopulation capacity as the leading Eigenvalue of a Jacobian matrix, you should say around which point the linearization is made.

In other words, according to Hanski and Ovaskainen in their Nature paper in 2000, metapopulation capacity should be greater than certain threshold for the whole metapopulation network to persist, but according to your presentation it is enough if the metapopulation capacity is greater than zero if the Jacobian corresponds to a time continuous model, or it is greater than 1 if the system is described as a time discrete Markov chain. I would like to know where this mismatch between Hanski-Ovaskainen interpretation of the metapopulation capacity and yours comes from. It would be really nice if you could clarify this point before publications.

Decision letter (RSOS-201309.R0)

Dear Dr Bertassello

On behalf of the Editors, we are pleased to inform you that your Manuscript RSOS-201309 "Dynamic Spatiotemporal Patterns of Metapopulation Occupancy in Patchy Habitats" has been accepted for publication in Royal Society Open Science subject to minor revision in accordance with the referees' reports. Please find the referees' comments along with any feedback from the Editors below my signature.

Please submit your revised manuscript and required files (see below) no later than 7 days from today's (ie 07-Dec-2020) date. Note: the ScholarOne system will 'lock' if submission of the revision is attempted 7 or more days after the deadline. If you do not think you will be able to meet this deadline please contact the editorial office immediately.

on behalf of Professor Tim Rogers (Associate Editor) and Pete Smith (Subject Editor)
openscience@royalsociety.org

Reviewer comments to Author:

Reviewer: 1
Comments to the Author(s)

The authors succeeded in doing a great job. I particularly like the description of patch networks by using parameters developed in the framework of network theory. I like very much how the hydrodynamic model is able to link rainfall to network structure in a dynamic way, which ends defining a constantly changing patch network. In sum, I love the combination of metapopulation ecology, network theory and hydrology.

My comment will be minor as I think the paper could be published almost as it is.

When you present the calculation the metapopulation capacity, I am a little bit worried about the interpretation of this parameter in the context of a rainfall-driven patch network. This parameter arises when trying to find a feasible non-trivial solution for the probability vector $p=(p_1, \dots, p_N)$, where p_i is the occupancy probability of the species on patch i . However, you say it is the leading Eigenvalue of the Jacobian of the system, but you don't present clearly enough the equations describing your dynamics. Could you specify the system of equations this Jacobian is representing and at which point is the Jacobian being evaluated? My worry is that if you defined the metapopulation capacity as the leading Eigenvalue of a Jacobian matrix, you should say around which point the linearization is made.

In other words, according to Hanski and Ovaskainen in their Nature paper in 2000, metapopulation capacity should be greater than certain threshold for the whole metapopulation network to persist, but according to your presentation it is enough if the metapopulation capacity is greater than zero if the Jacobian corresponds to a time continuous model, or it is greater than 1 if the system is described as a time discrete Markov chain. I would like to know where this mismatch between Hanski-Ovaskainen interpretation of

the metapopulation capacity and yours comes from. It would be really nice if you could clarify this point before publications.

===PREPARING YOUR MANUSCRIPT===

===PREPARING YOUR REVISION IN SCHOLARONE===

- 1) One version identifying all the changes that have been made (for instance, in coloured highlight, in bold text, or tracked changes);
 - 2) A 'clean' version of the new manuscript that incorporates the changes made, but does not highlight them.
 - An individual file of each figure (EPS or print-quality PDF preferred [either format should be produced directly from original creation package], or original software format).
 - An editable file of each table (.doc, .docx, .xls, .xlsx, or .csv).
 - An editable file of all figure and table captions.
- Note: you may upload the figure, table, and caption files in a single Zip folder.
- Any electronic supplementary material (ESM).
 - If you are requesting a discretionary waiver for the article processing charge, the waiver form must be included at this step.
 - If you are providing image files for potential cover images, please upload these at this step, and inform the editorial office you have done so. You must hold the copyright to any image provided.
 - A copy of your point-by-point response to referees and Editors. This will expedite the preparation of your proof.

- Ensure that your data access statement meets the requirements at <https://royalsociety.org/journals/authors/author-guidelines/#data>. You should ensure that you cite the dataset in your reference list. If you have deposited data etc in the Dryad repository, please only include the 'For publication' link at this stage. You should remove the 'For review' link.
- If you are requesting an article processing charge waiver, you must select the relevant waiver option (if requesting a discretionary waiver, the form should have been uploaded at Step 3 'File upload' above).
- If you have uploaded ESM files, please ensure you follow the guidance at <https://royalsociety.org/journals/authors/author-guidelines/#supplementary-material> to include a suitable title and informative caption. An example of appropriate titling and captioning may be found at [https://figshare.com/articles/Table_S2_from_Is_there_a_trade-off_between_peak_performance_and_performance_breadth_across_temperatures_for_aerobic_sc ope_in_teleost_fishes_/3843624](https://figshare.com/articles/Table_S2_from_Is_there_a_trade-off_between_peak_performance_and_performance_breadth_across_temperatures_for_aerobic_scope_in_teleost_fishes_/3843624).

Author's Response to Decision Letter for (RSOS-201309.R0)

See Appendix A.

Decision letter (RSOS-201309.R1)

This year has been very difficult for everyone, and we want to take the opportunity to thank you for your continued support in 2020.

The Royal Society Open Science editorial office will be closed from the evening of Friday 18 December 2020 until Monday 4 January 2021. We will not be responding during this time. If you have received a deadline within this time period, please contact us as soon as possible to allow us to extend the deadline. If you receive any automated messages during this time asking you to meet a deadline, we offer apologies and invite you to respond after the festive period or during normal working hours.

With our best for a peaceful festive period and New Year, and we look forward to working with you in 2021.

Dear Dr Bertassello,

It is a pleasure to accept your manuscript entitled "Dynamic Spatiotemporal Patterns of Metapopulation Occupancy in Patchy Habitats" in its current form for publication in Royal Society Open Science.

on behalf of Professor Tim Rogers (Associate Editor) and Pete Smith (Subject Editor)
openscience@royalsociety.org

Appendix A

Reviewer: 1

Comments to the Author(s)

R1C1: The authors succeeded in doing a great job. I particularly like the description of patch networks by using parameters developed in the framework of network theory. I like very much how the hydrodynamic model is able to link rainfall to network structure in a dynamic way, which ends defining a constantly changing patch network. In sum, I love the combination of metapopulation ecology, network theory and hydrology.

R: The authors wish to thank this Reviewer for her/his appreciation of our work.

My comment will be minor as I think the paper could be published almost as it is.

R1C2: When you present the calculation the metapopulation capacity, I am a little bit worried about the interpretation of this parameter in the context of a rainfall-driven patch network. This parameter arises when trying to find a feasible non-trivial solution for the probability vector $p=(p_1, \dots, p_N)$, where p_i is the occupancy probability of the species on patch i . However, you say it is the leading Eigenvalue of the Jacobian of the system, but you don't present clearly enough the equations describing your dynamics. Could you specify the system of equations this Jacobian is representing and at which point is the Jacobian being evaluated? My worry is that if you defined the metapopulation capacity as the leading Eigenvalue of a Jacobian matrix, you should say around which point the linearization is made.

R: We believe that there has been some misinterpretation, likely generated by our text that was overly summarized on this point. Indeed, please note that the metapopulation capacity λ_{\max} is not computed as the leading eigenvalue of the Jacobian matrix J of the system, as this Reviewer concluded at the end of this comment, but rather as the leading eigenvalue of the landscape matrix M consisting of elements $m_{ij} = \exp(-d_{ij}/D)S_iS_j$ for $i \neq j$ and $m_{ii} = 0$, both in the static and dynamic model. We have expanded this paragraph on the revised version of the text to clarify this point (Lines: 185-194). The novelty introduced in our paper is that the interpatch distance d_{ij} , and the size of the patches S_i and S_j change in time in response to the hydroclimatic variability.

R1C3: In other words, according to Hanski and Ovaskainen in their Nature paper in 2000, metapopulation capacity should be greater than certain threshold for the whole metapopulation network to persist, but according to your presentation it is enough if the metapopulation capacity is greater than zero if the Jacobian corresponds to a time continuous model, or it is greater than 1 if the system is described as a time discrete Markov chain. I would like to know where this mismatch between Hanski-Ovaskainen interpretation of the metapopulation capacity and yours comes from. It would be really nice if you could clarify this point before publications.

R: As we pointed out in the previous response to the comment, there is no theoretical mismatch between ours and Hanski-Ovaskainen interpretation of the metapopulation capacity. The main difference is that in our approach the metapopulation capacity λ_{\max} is not constant but varies as the size of the wetlands and their inter-patch distance change. We are confident that the revised manuscript clarifies this point (Lines 195-203) and we wish to thank this Reviewer for pointing out this possible source of misinterpretation.